# Performance and Wear of Diamond Honing Stones

Georg Mahlfeld * and Klaus Dröder

Institute of Machine Tools and Production Technology, Technische Universität Braunschweig, Langer Kamp 19b, 38106 Braunschweig, Germany
* Correspondence: g.mahlfeld@tu-braunschweig.de; Tel.: +49-531-391-7699

**Abstract:** Honing is one of the most precise processes for the production of tribologically highly loaded cylindrical bores in different areas of technology. The honing stone specifications influence the process performance to a large extent. During machining, the honing stones are in permanent contact with the workpiece and subject to high mechanical loads. High strength steel and latest coating materials cause additional stress on the honing stones and induce increased wear. The objective is to determine process information for these material properties to support an effective process design. In experiments, the performance and the wear rates of several honing stone specifications were investigated. In this publication, the observed wear mechanisms are analysed and influencing factors for reduced wear are outlined.

**Keywords:** honing; honing stone; tool wear; diamond grain

## 1. Introduction

Honing has a multitude of applications in various industrial sectors, as it serves as a finishing process for a wide range of workpieces. Examples, apart from piston–cylinder pairings, can be found in hydraulics and pneumatics, shock absorbers, transmissions, pumps and compressors [1,2]. For linear guides and plain bearings, honed surfaces also ensure low friction and a high durability. For all honing processes, the honing stone is the key interface between machine and workpiece. In the machine–tool–workpiece functional relationship, the honing stone specification determines the performance and the result of the entire process. In most applications, the technical requirements cannot be achieved with a single honing step and one honing stone specification. For this reason, several honing steps with graded specifications are used categorised in prehoning, intermediate honing and finishing specifications. The honing stone structure is a composite of cutting grains embedded in a bond material. The cutting materials are divided into high-hardness and conventional cutting materials. High-hardness cutting materials include industrial diamond and boron nitride, while conventional cutting materials are generally silicon carbide and aluminium oxide [3]. With 95% of all applications, the high-hardness cutting materials are mainly used in honing. With a total of 65%, industrial diamond is the most widely applied cutting material in honing and has the highest technical relevance [3]. In order to achieve sufficiently long tool lives there is no alternative to industrial diamond as a cutting material for finishing high-strength steels in mechanical engineering and latest thermal spray coatings.

Previous research has investigated the wear and stability criteria of diamond grains, the influence of the bonding material and the process parameters. Furthermore, various approaches for optimisation and evaluation have been developed [4]. With same properties small grains mainly fail due to grain pull out whereas big grains predominantly tend to grain fracture. In scratch tests for hardened 15NiCr 13, a significant effect of the rake angle on the cutting forces is observed. The orientation and geometry of the grains therefore significantly influences the cutting forces and thus the probability of failure [5].

An evaluation of common metallic bonded diamond honing stones in comparison with vitrified and resinoid bonds for grey cast iron was performed by Sabri et al. The quality at the form scale (cylindricity) is reduced and the quality at roughness scale is improved for diamond resinoid stones. Vitrified and resinoid bonds show a high finishing performance with a limitation on wear. The highest tool life is achieved for metallic bonds [6]. Major influences from the process on the stone wear can be derived from the cutting speed, the contact pressure and the feed strategy. All investigations demonstrate a disproportionate increase in wear for higher cutting speeds and a higher contact pressure compared to the increased metal removal rate [7,8]. Compared to feed-controlled strategies a force-controlled approach increases the process stability, the quality of the honed workpieces and the tool life [1]. An investigation on roughness variations by honing tool wear for grey cast lines was performed by Cabanettes et al. It is shown that the series production process is very reliable and that only minor changes for the roughness peaks of the surface profile can be detected over the tool life of the honing stones [9]. Most studies focus on grey cast iron as a material. Some studies also consider hardened steel or ceramic materials. Thermal spray coatings and materials with equivalent properties have not been sufficiently researched.

The goal of this research is to investigate the wear rate and the dominating mechanisms for diamond grain honing stones for latest liner materials in order to support effective process design. Experiments were carried out in pre, intermediate and finish honing of a common heat-treated steel 100Cr6 (36 HRC), which corresponds to the material properties of latest thermal spray coatings [10]. Eight honing steps with grain sizes D181 to D7 are examined, as this range is practically relevant for many applications in mechanical and automotive engineering. With this systematic investigation, a comprehensive database is collected in order to analyse the process and enable potential improvements. The results show the machining quality on the workpiece, the cutting performance and the measured wear. The wear mechanisms for the specifications used were analysed in detail using SEM images (scanning electron microscope) and innovative computer tomography.

## 2. Types and Characteristic Values of Honing Stone Wear

Honing stones are a composite system combining a base for mounting and a cutting layer for machining. The cutting layer consists of bond material and embedded grains. Based on this structure, the wear of the honing stone occurs in the form of grain and bond wear (see Figure 1). The honing stone properties are specifically adjusted to the workpiece material characteristics in order to obtain a consistently balanced wear of the cutting grains and the bond material. Dulled grains should break out of the bond when a force threshold is exceeded, and a parallel resetting of the bond is required in order to expose new grains. This balance is called self-sharpening of honing stones. During machining, the cutting layer is subject to progressive wear until the layer is completely worn and the honing stones need to be replaced.

Grain wear appears as three types: abrasion (dulling), splintering and break-out of whole grains. The proportions of these wear types are determined by the properties of the grains used. The grains of high-hardness cutting materials differ not only in size, but also in shape, structure and physical properties. In contrast to natural diamonds, which continue to be less and less important for honing, synthetic industrial diamonds can be modified during the manufacturing process and thus adapted to the machining task [5].

The grain quality of industrial diamond is divided into two general categories, monocrystalline and multicrystalline grains. Monocrystalline grains have a hexahedral or dodecahedron shape and are geometrically more perfect, the lower the degree of impurity. Multicrystalline grains are endowed in a synthesis process to grow different crystal faces. Compared to multicrystalline diamonds, monocrystalline diamonds have a higher hardness and are therefore more wear resistant. Typically, splintering does not occur for monocrystalline grains in honing. As a result, the machining process causes the grains to be dulled over time until the cutting forces exceed the grain holding force in the bond. The monocrystalline grains are mainly used for rough honing. In this honing step, a high

metal removal rate is desired. The surface quality is of secondary interest as it can be improved with further honing steps. Multicrystalline diamonds are characterised by a high splintering ability, which means that new and sharp micro edges are repeatedly exposed in the machining process. These grains are therefore mainly used for finishing, as the generated micro edges achieve a better surface roughness. The difference between the grain types and the resulting wear progression is shown in Figure 2.

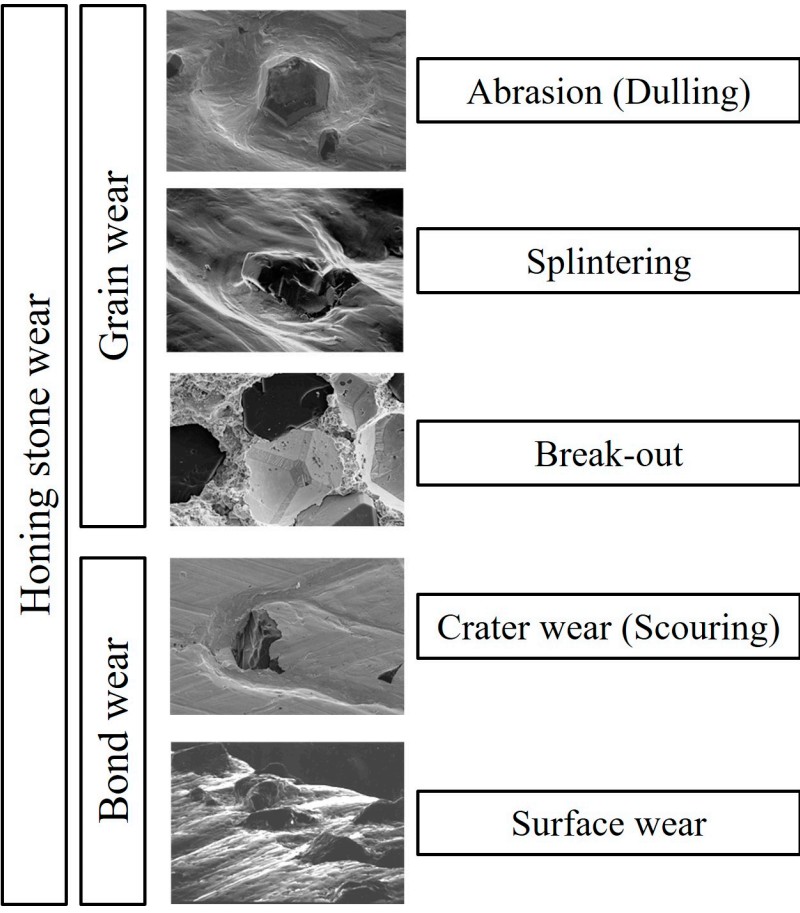

**Figure 1.** Types of honing stone wear, according to [3].

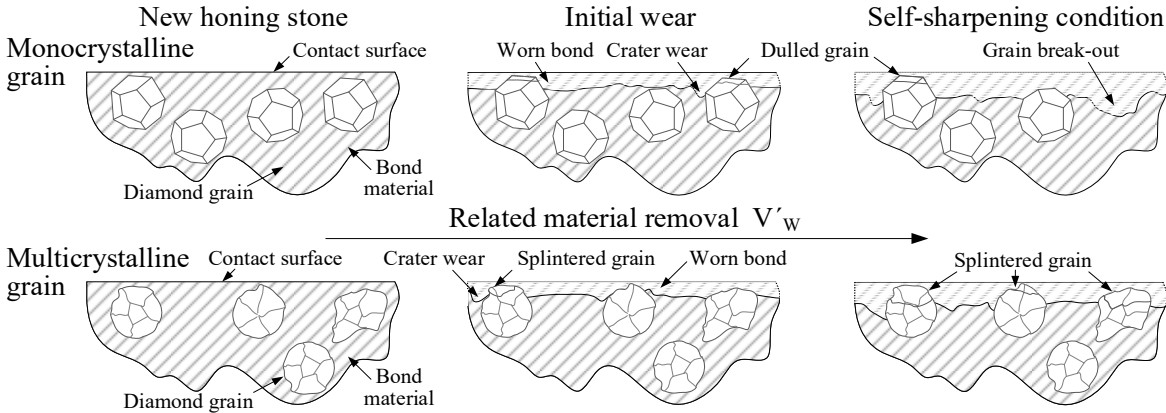

**Figure 2.** Schematic of honing stone wear for monocrystalline and multicrystalline diamond grains.

The bond represents the main volume component within the cutting layer of a honing stone and has the function of holding the cutting grains in place until the cutting perfor-

mance is insufficient by dulling. Afterwards, the dulled grains are supposed to disconnect form the bond in order to obtain subsequent sharp grains in operation.

Bond wear appears as two types, crater wear and surface wear. The surface wear becomes visible as grooves in the honing stone surface and is caused by the contact between bond and workpiece surface and their relative movement. The total wear of the honing stone is significantly influenced by the bond wear. To a large extent, this depends on the mechanical properties of the bond material. An additional factor influencing bond wear is the grain concentration (see Figure 3). With an increase in grain concentration, the contact between the workpiece and the honing stone is distributed over a higher number of grain tips. At the same time, a higher grain concentration reduces the volume share of bond material and causes a stronger influence of the grains on each other. Consequently, a higher concentration does not necessarily result in a higher material removal rate or lower honing stone wear as the data from [11] indicate.

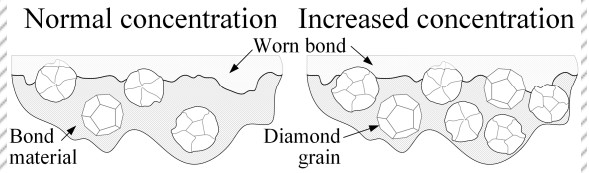

**Figure 3.** Schematic for variations in grain concentration.

In the gap between the honing stone and the workpiece the bonded grains, the bond, the chips, the workpiece surface as well as the loose or splintered grains collide and influence each other. The geometric interference of the bonded grains with the workpiece surface leads to the intended workpiece machining. Honing is a force dependent process, as the interference and thus the chip thickness is a result of the applied normal force $F_N$. The resulting chips rub against the bond and the workpiece when the honing oil rinses them out (see Figure 4). Chip friction primarily induces wear on the bond, as the bond generally has a lower hardness than the workpiece. This wear appears on the honing stone as crater wear directly in front of the grain and behind the grain as grooves oriented in honing angle α.

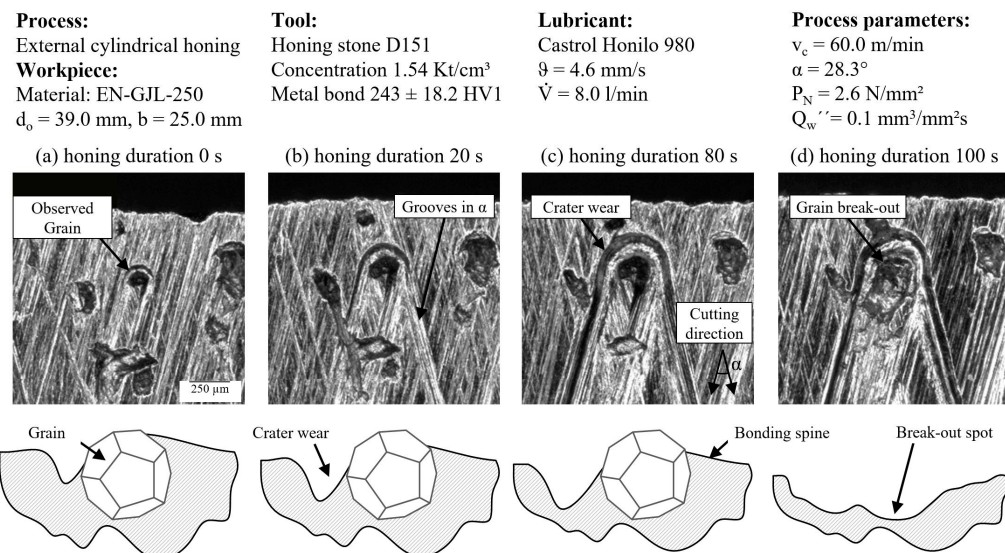

**Figure 4.** Progress of crater wear until grain break-out. (The experimental setup is as described in Section 4.)

### 3. Wear Values and Coefficients

Previous research used various methods to specify tool wear. In honing, common characteristic values and coefficients are the wear per honing cycle [9], the wear gradient [11,12] and the G-Factor [2,7]. The wear per honing cycle is a value specific for the investigated process. The value allows a quick comparison of the wear for different honing stone specifications or changed process parameters for a specific experimental setup. Another advantage is that the number of components until tool change can be calculated directly from the wear per honing cycle and the thickness of the honing stone used. However, this characteristic value does not allow a comparison across different honing processes, as honing stone width, number of honing stones, workpiece properties and process time are not considered.

The description of the wear by means of a gradient or a wear rate per second includes the honing time to improve comparability. Nevertheless, these related values are only useful in direct comparison with data from the same experimental environment. For this reason, the G-Factor also referred to as grinding ratio is often used. Originally, the G-Factor is applied to conventional abrasives such as aluminium oxide. The G-Factor is the amount of machined material per unit wear volume, which is a comparable key figure. The consideration of the material removal includes the machining time; thus, the G-Factor is not influenced by the honing time. The value allows for a quick comparison of the wear for different stone specifications or changed process parameters for a repeatable test setup. However, with varying processes large differences can occur. For this reason, the factor should not be considered in isolation and at least the stock removal volume should be included for comparison.

The G-Factor is based on the wear volume of the honing stones and not on the wear volume of the actual cutting grain. Due to the structure of diamond honing stones, which is different in volume fractions from conventional abrasives, the concentration of grains $C$ ($Kt/mm^3$) has an influence on the G-Factor. For diamond honing stones, the focus should be on the consumption of grains instead of considering the wear volume exclusively. In the production of honing stones, the diamond grains cause the major costs. For this reason, the additional declaration with the characteristic value grain utilisation factor (U-Factor) is proposed.

The calculation of the U-Factor is an extension of the calculation of the G-Factor. The volume of machined material $V_M$ ($mm^3$) is related to the weight of grains, calculated from the concentration $C$ ($Kt/mm^3$) and the worn honing stone volume $V_W$ ($mm^3$). For better comprehensibility, the unit carat (Kt) is converted to the SI unit gram (g) (1 Kt = 0.2 g).

$$\text{Grain utilisation factor} = \frac{V_M}{0.2 \times C \times V_W} \ mm^3/g \qquad (1)$$

The U-Factor as an additional key figure expresses the grade of economic cutting material usage for interpretation of the honing stone performance. The characteristic value allows a comparison of different honing stone specifications and can be used in conjunction with the material removal rate for iterative process optimization. At the same time, the comparability of different honing processes with each other is improved compared to the state-of-the-art values.

A comparison of different honing processes from the literature illustrates the correlations between the key figures (see Table 1). While [2,12] illustrate analogous processes on an experimental scale, [11] imitates a high productive bore machining process from the automotive industry.

In summary, a single characteristic value is not sufficient for specifying honing stone wear. Assuming a complete description of the applied process, the tools and the workpieces, it is recommended to use the U-Factor to quantify diamond honing stone wear.

**Table 1.** Honing stone wear values collected from literature.

| Source | [12] | [2] | [11] |
|---|---|---|---|
| Specification | D181 | D64 | D25 |
| Specific material removal rate (mm$^3$/mm$^2$·s) | 0.58 | 0.006 | 0.016 |
| Wear per honing cycle (mm$^3$) | 12.0 | 0.69 | 8.25 |
| Wear gradient (mm$^3$/s) | 0.0134 | 0.0116 | 0.4124 |
| G-Factor | 437.5 | 45 | 74 |
| Grain concentration C (Kt/mm$^3$) | 2.64 | 2.20 | 1.54 |
| U-Factor (mm$^3$/g) | 828.6 | 102.25 | 240.25 |

## 4. Experimental Setup and Machining Conditions

To quantify the wear type contribution to the total honing stone wear, experiments were carried out performed on a test rig for external cylindrical honing similar to [2,13]. The workpiece was mounted on a spindle for rotational movement. The normal force $F_N$ is applied via a hydraulic drive in order to press the rotating workpiece against an oscillating honing stone attached to a hydraulically driven oscillating table.

For the experiments, ring-shaped workpieces of bearing steel 100Cr6 were used. This steel represents a standard material for friction loaded components. For a reasonable comparison to latest thermal spray layers used in automotive applications the rings are heat-treated to 364 ± 12 HV 0.3. The rings were $d_0$ = 50 mm in diameter and b = 30 mm in width prepared by turning to a similar initial roughness of Rz = 10 μm.

Diamond honing stones with sintered metal bond were used for the experiments (see Table 2). Three specifications for prehoning (D181, 107, D91), two for intermediate honing (D64, D46) and three for finishing (D25, D15, D7) were examined. The dimensions were a length of l = 20 mm and a height of h = 3 mm resulting in a contact surface of $A_C$ = 60 mm$^2$. Before the experiment, the honing stones were prepared by running-in on the test rig to the workpiece radius in order to achieve full contact and a steady state condition of the honing stone topography.

**Table 2.** Honing stone specifications used in experiments.

| | Specification | Crystal Type | Bond Material | Bond Hardness | Grain Concentration C (Kt/mm$^3$) |
|---|---|---|---|---|---|
| Prehoning | D181 | Monocrystalline | W-Co alloy | 337 ± 37.5 | 2.2 |
| | D107 | Monocrystalline | Fe alloy | 287 ± 16.7 | 2.2 |
| | D91 | Monocrystalline | W-Co alloy | 337 ± 37.5 | 2.2 |
| Intermediate | D64 | Blend of mono- and multicrystalline | Fe alloy | 287 ± 16.7 | 2.2 |
| | D46 | Blend of mono- and multicrystalline | Fe alloy | 265 ± 18.3 | 1.5 |
| Finishing | D25 | Multicrystalline | Fe alloy | 265 ± 18.3 | 1.5 |
| | D15 | Multicrystalline | Bronze alloy | 184 ± 22.3 | 1.1 |
| | D7 | Multicrystalline | Bronze alloy | 184 ± 22.3 | 1.1 |

## 5. Performance and Grain Utilisation in Honing 100Cr6 Steel

With these honing stones, the following surface roughnesses were achieved (see Table 3). The values were calculated from five independent tactile measurements carried out on the workpiece surface according to DIN EN ISO 27178. The robust profile filter Gaussian regression filter according to DIN EN ISO 16610-31 was used because of the

characteristic honing surface with grooves and impulsive disturbances. The material content curve (Abbott) and the characteristics Rk, Rpk and Rvk were calculated as per DIN EN ISO 13565-2. The measured values are provided to ensure full traceability of the experimental results. The subsequent analysis of the process performance is carried out based on the average roughness depth Rz.

**Table 3.** Results of the surface roughness for the used honing stones.

| Specification | Ra (µm) | Rz (µm) | Rk (µm) | Rpk (µm) | Rvk (µm) |
|---|---|---|---|---|---|
| D181 | 2.59 | 22.82 | 7.71 | 3.43 | 5.23 |
| D107 | 2.32 | 19.55 | 7.32 | 3.09 | 4.56 |
| D91 | 2.27 | 18.89 | 6.90 | 2.55 | 4.33 |
| D64 | 1.29 | 9.40 | 3.71 | 1.72 | 2.25 |
| D46 | 1.17 | 8.85 | 3.54 | 1.62 | 2.04 |
| D25 | 0.29 | 2.43 | 0.88 | 0.44 | 0.40 |
| D15 | 0.19 | 1.73 | 0.57 | 0.26 | 0.29 |
| D7 | 0.06 | 0.82 | 0.10 | 0.05 | 0.28 |

In prehoning small differences in surface roughness were measured related to the difference in grain size. For D91 with 50% grain diameter compared to D181 the average roughness value Ra is reduced by 14%. Due to the relatively small differences in surface roughness, multiple prehoning steps do not contribute to reduced lead time. Similar to prehoning, minor differences in surface quality were measured in intermediate honing. Therefore, a single intermediate honing step is sufficient for an efficient process chain. Finishing with grain sizes D25, D15 and D7, a significant reduction in surface roughness was achieved compared to the previous specifications. The main reason are the microcrystalline diamond grains, which form micro-edges by splintering and thus penetrate the material surface less deep. In the specifications investigated for finishing, a significant reduction in the average roughness depth from Rz (D25) = 2.4 µm and Rz (D15) = 1.7 µm to Rz (D7) = 0.8 µm were measured with decreasing grain size. Due to these major differences in finishing, two honing stages should be considered for highest qualities. With the D7 specification, exclusively the roughness peaks are machined. This is demonstrated by a lower reduced peak height Rpk and reduced core depth Rk. An improvement of the reduced groove depth is not possible with a D7 honing stone. In comparison with other investigations, higher surface roughnesses are produced in the experiments. The main reason for this is a higher contact pressure on the honing stone compared to other studies [12,14].

The analyses of the honing stone wear (G-Factor, U-Factor), the material removal rate ($Q_w$) and the surface roughness (Rz) are shown in Figure 5. The results are plotted over the used grain size. The experiments were carried out starting from the largest and proceeding to the smallest grain size. With this method, an experiment starts with the surface roughness of the next larger grain size and the resulting surface roughness is used for the next smaller grain size. The given values are the average values from ten individual experiments performed on ten workpieces.

Based on the measured roughness values, the honing stone specifications for prehoning, intermediate honing and finish honing can be clearly identified. The preliminary honing operations are more relevant in terms of cycle times due to higher stock removal. With the honing stones for prehoning with grain sizes D181, D107 and D91 the highest material removal rate was measured. Accordingly, a high surface roughness for these honing stones between Rz = 22.8 and Rz = 18.9 µm was observed. Despite the high cutting performance in prehoning, the highest G-Factor was achieved. The reason is the high wear resistance of the monocrystalline grains combined with a high bond hardness. As seen in the magnification in Figure 6, no splintered grains, but a high number of grains with wear marks are visible after machining. The signs of dulling and micro-fractures indicate a long engagement time. It can therefore be expected that the grains will not be overloaded by

the cutting forces. In front of the grains crater wear and distinct bonding spines (leftover bonding material) behind can be observed for all operating grains. This indicates a strong adhesion of the grains to the bond and low stress on single grains resulting in a long operation time.

**Process:**
External cylindrical honing
**Workpiece:**
Material: 100Cr6 (1.2067)
$d_o$ = 50.0 mm, b = 30.0 mm

**Tool:**
Diamond honing stone
Concentration:
2.2 - 1.1 Kt/cm³
Metal bond

**Lubricant:**
Castrol Honilo 980
$\vartheta$ = 4.6 mm/s
$\dot{V}$ = 8.0 l/min

**Process parameters:**
$v_c$ = 60.0 m/min
$\alpha$ = 39.6°
$P_N$ = 2.3 N/mm²

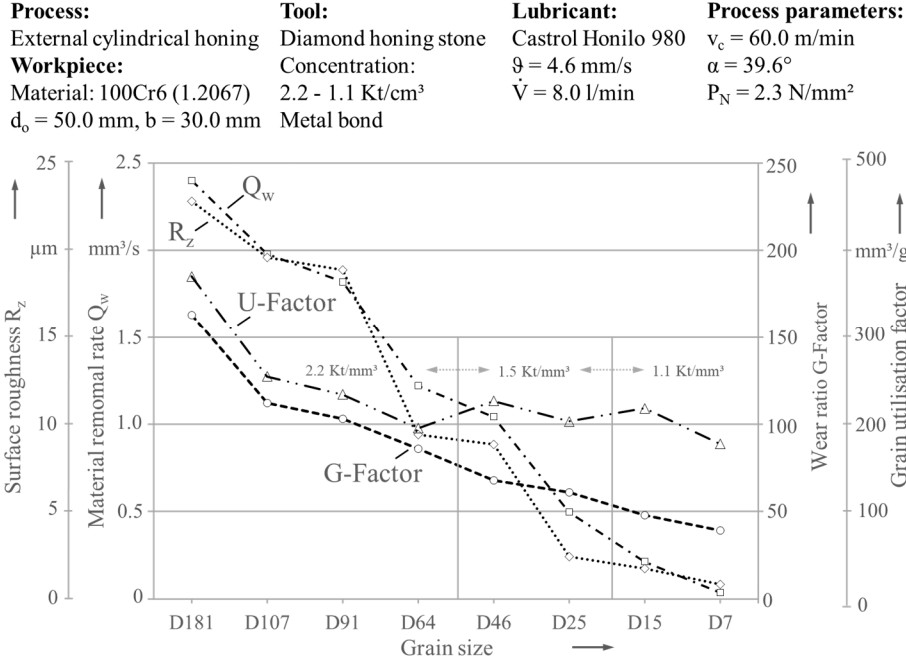

**Figure 5.** Cutting performance and wear of diamond honing stones.

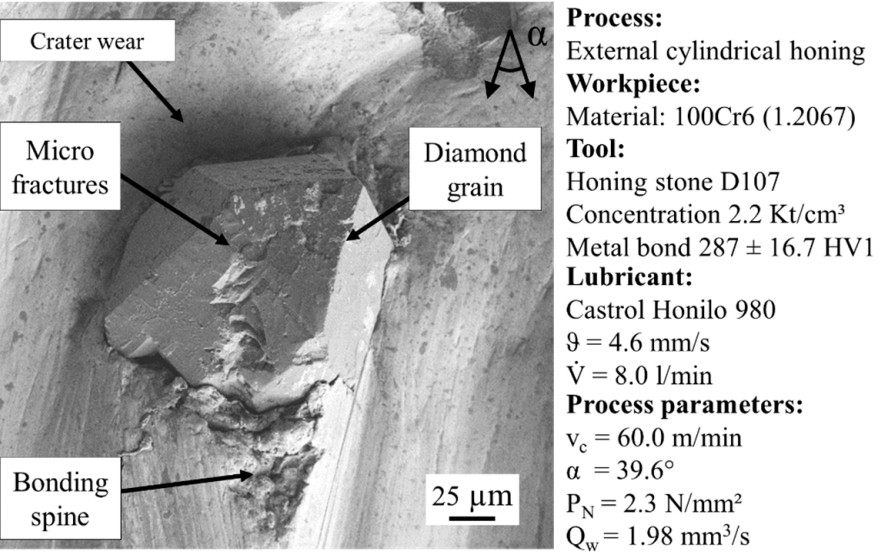

**Process:**
External cylindrical honing
**Workpiece:**
Material: 100Cr6 (1.2067)
**Tool:**
Honing stone D107
Concentration 2.2 Kt/cm³
Metal bond 287 ± 16.7 HV1
**Lubricant:**
Castrol Honilo 980
$\vartheta$ = 4.6 mm/s
$\dot{V}$ = 8.0 l/min
**Process parameters:**
$v_c$ = 60.0 m/min
$\alpha$ = 39.6°
$P_N$ = 2.3 N/mm²
$Q_w$ = 1.98 mm³/s

**Figure 6.** Wear marks for monocrystalline grains on prehoning.

For the intermediate operations, the grain sizes D64 and D46 were used. In comparison to all specifications examined, medium removal rates and equivalent medium roughness values were measured. Due to the combination of mono- and multicrystalline grains, the wear ratio is reduced compared to the prehoning specifications. On the honing stone surface, according to the grain combination, splintered grains and bound grains with abrasion marks can be observed (see Figure 7). The crater wear for splintered grains is significantly smaller than for monocrystalline grains after a long engagement time. This indicates smaller chips and lower friction on the bond induced by less intrusion depth. Wear

has slightly increased with a G-Factor of 85.9 for D64 and 68.2 for the D46 specification. Due to a reduced grain concentration of C = 1.5 Kt/mm$^3$ for D46, the U-Factor does not differ from the prehoning stones. Therefore, with reduced grain size, reduced grain concentration has a positive effect on grain utilisation.

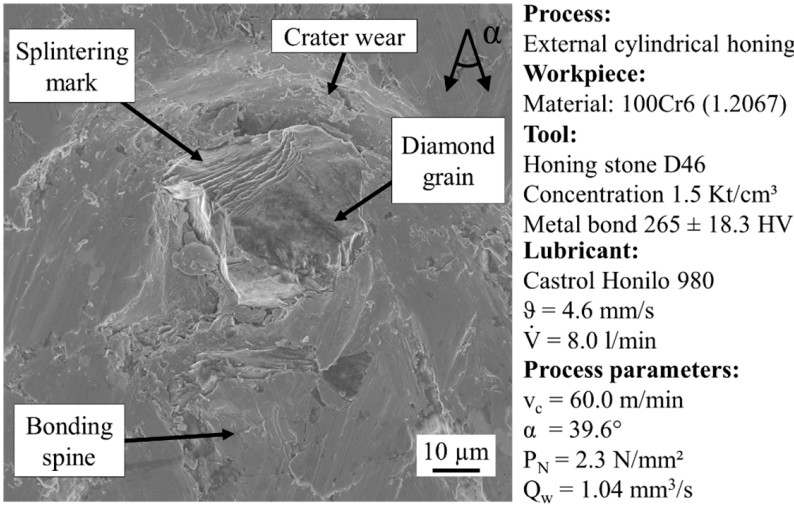

**Process:**
External cylindrical honing
**Workpiece:**
Material: 100Cr6 (1.2067)
**Tool:**
Honing stone D46
Concentration 1.5 Kt/cm³
Metal bond 265 ± 18.3 HV1
**Lubricant:**
Castrol Honilo 980
$\vartheta$ = 4.6 mm/s
$\dot{V}$ = 8.0 l/min
**Process parameters:**
$v_c$ = 60.0 m/min
$\alpha$ = 39.6°
$P_N$ = 2.3 N/mm²
$Q_w$ = 1.04 mm³/s

**Figure 7.** Wear marks for intermediate honing.

In finish honing with the grain sizes D25, D15 and D7, a significant reduction in the material removal rate and the surface roughness was achieved. Grain wear occurred exclusively by splintering as expected for multicrystalline grains. The splintering of the grains creates new micro edges and reduces the overall grain size. This reduces the grain intrusion depth and therefore the machined surface roughness. The comparison between G-Factor and U-Factor indicate a positive effect of lower grain concentration on grain utilisation. The reason is that a lower concentration and thus number of grains reduce the influence of the particles on each other in the gap. Thus, a lower concentration results in a lower probability that splintered cutting material in the gap will hit another grain and induces premature wear. At the same time, the probability that grains have a lower adhesion to the bond caused by the contact to another grain is reduced.

On the one hand, the test results show the influence of the honing stone specification on the machining result and the honing stone wear. On the other hand, the comparison of the key figures used show that the U-Factor facilitates a comparison across different specifications and is thus an additional evaluation opportunity.

## 6. Observed Wear Mechanisms

In addition to the types of honing stone wear and the mechanisms of previous investigations described in Section 2, further influences were observed. On the honing stone surface next to the grooves caused by chip friction, deeper marks are present. These originate from the interaction of grains and particles in the gap between the honing stone and the workpiece. Grains that have been broken out and move through the gap due to the relative movement induce grooves in the bond. Since the gap is smaller than the grain size and the bond hardness is lower than the workpiece hardness, the bond is primary machined. The induced grooves characteristically start at the grain breakout point and run along a line in honing angle direction to the end of the honing stone (See Figure 8).

Due to the high number of active grains on the honing stone surface, collisions of emitted grains also occur with bound grains (see Figure 9). In the case illustrated, a monocrystalline grain broke out of the bond and is wedged between workpiece and honing stone. Due to the relative movement, the grain is driven out of the gap. In this process, the closest grain was damaged. As a result, the grain splintered, which is atypical for monocrystalline specifications. Therefore, it can be derived that collisions induce

premature splintering or breaking out and thus increase the wear rate. This is relevant for monocrystalline specifications since these grains do not splinter under pressure and thus do not reduce the size. These grains therefore cause greater grooves in the bond and a higher probability of grain breakout in case of collision.

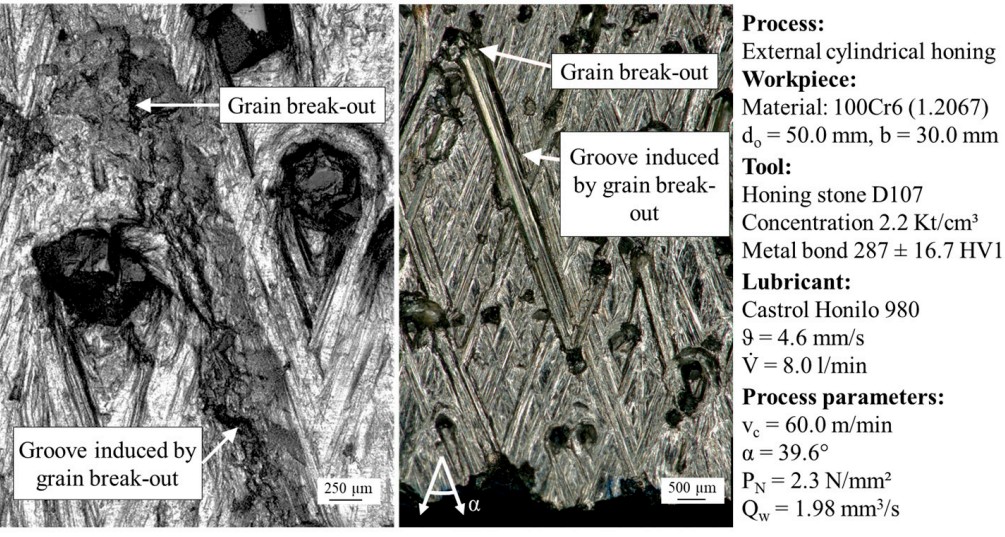

**Figure 8.** Bond wear in form of grooves induced by grain break-out.

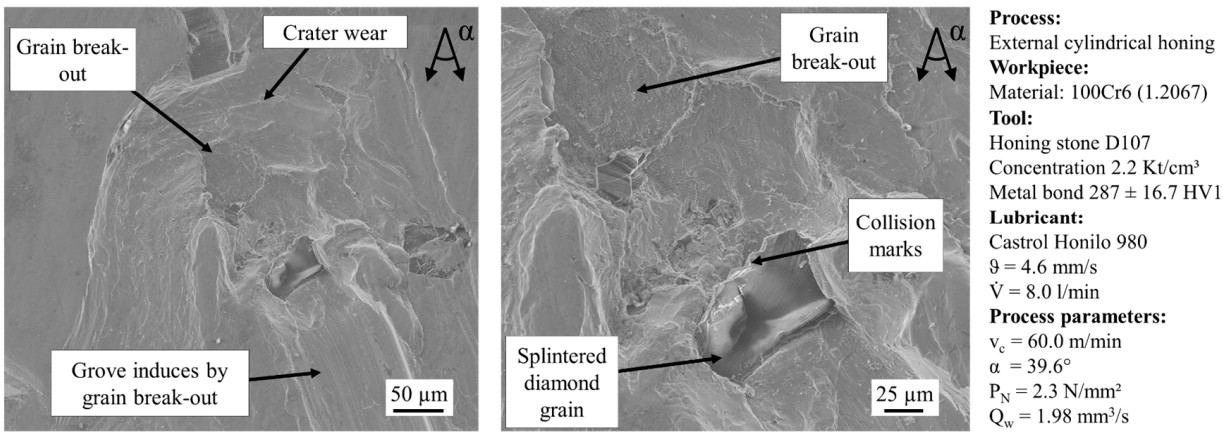

**Figure 9.** SEM images of grain collisions.

Nevertheless, these effects are also important for multicrystalline specifications since the gap is narrower and thus the splintered grain components already induce bond abrasion. Grain concentration has a high influence on this mechanism. The chipping of the bond by loose grains at a higher concentration means that an increased wear of the bond is caused despite the fact that the bond already accounts for a lower volume share. The results on grain utilisation show a lower wear with a reduction in grain concentration due to a decreased collision probability. At the same time, a narrow honing stone width is recommended in order to reduce the distance of interaction of loose grains and other particles in the gap.

During the analyses of wear mechanisms by SME images grain accumulations on the honing stone surface were observed. Subsequently, an investigation to measure the distribution of the grains within the cutting layer was carried out. Using computer tomography, a digital 3D volume replica of the honing stone was made. From this digital replica, sectional views of the volume were generated and analysed (see Figure 10). The evaluation of the images allocates the grains to a grid based on the centre point. The average number of grains per grid can be determined via the dispersion of the grains within the grid fields.

The arithmetic average value x̄ is calculated afterwards. The quality of the dispersion is determined by the variance $\sigma^2$ and the standard deviation $\sigma$. In the case of a homogeneous dispersion of the cutting grains in the cutting layer, the same number of grains would be present in all grid areas.

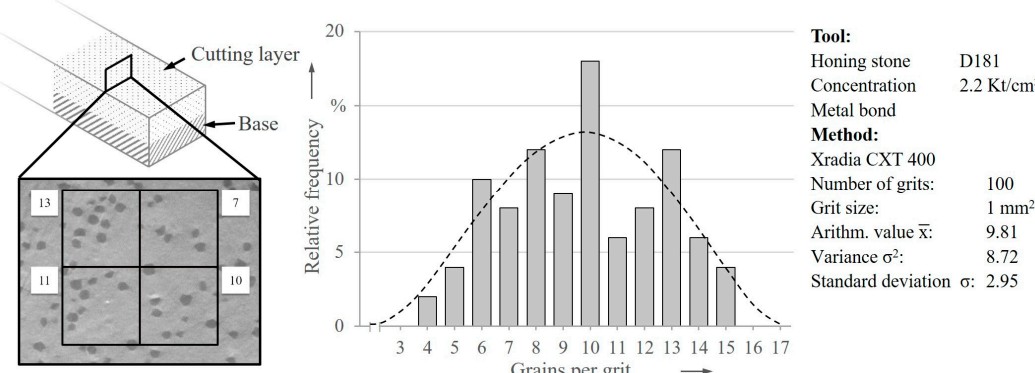

**Figure 10.** Distribution quality examined by computed tomography.

The average number of grains per grid is x̄ = 9.81, which is the expected value for the given specification. The dispersion of grains shows strong inequalities with a variance of $\sigma^2$ = 8.72 and a standard deviation of $\sigma$ = 2.95. A general displacement of the grains to the edges or the honing stone centre was not observed. The unequal dispersion of the grains has a negative effect on the cutting performance and wear. This can be seen on the cross-section of the honing stone (Figure 11).

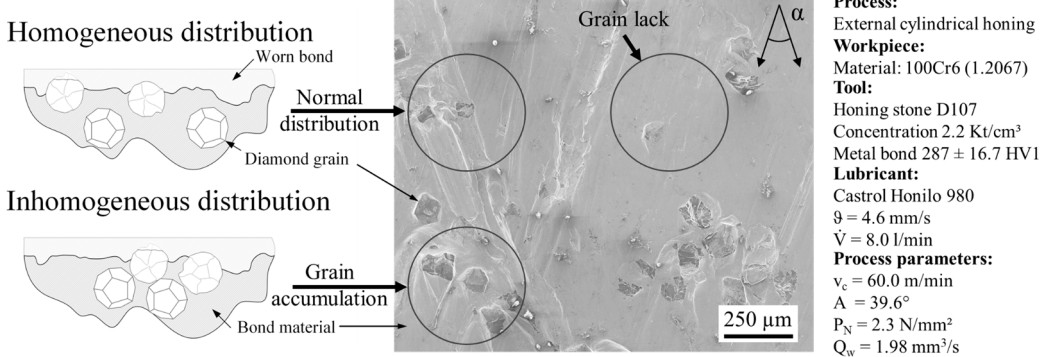

**Figure 11.** Schematic sketch for variations in grain distribution.

The accumulation of grains increases the probability of negative interaction between the grains. A grain breakout is therefore more likely to cause damage to another grain or reduce the bond strength of further grains. This is illustrated in Figure 11 by the fact that the crater wear of one grain overlaps with the surrounding bond of another grain. In areas with accumulations a faster bond wear occurs. As a result, the bonding forces of individual grains are reduced. In areas with a low grain number, the bond resets more slowly, as shown in the upper section of Figure 11. These areas with grain lack cause friction between the bond and the workpiece for bond reset. As a result, the amount of contact force $F_N$ for cutting is reduced and thus decreases the interference of grains with the workpiece. Conversely, a lower cutting performance is achieved. An equal dispersion of grains in the entire cutting layer should therefore be the prospective objective of honing stone production.

The results of these observations show that in honing stone wear analyses, two additional wear influences must be considered. First is the bond wear induced by grain break-out. Second the grain interaction in the gap between honing stone and workpiece.

Both factors must be taken into account when developing honing stone specifications in order to achieve maximum performance and minimum wear. For the first time, an inhomogeneous distribution of the grains in the cutting layer has been demonstrated. Here, it is expected that a homogeneous distribution will reduce wear and improve the machining performance.

## 7. Conclusions

The composition of a honing stone essentially determines the machining performance, surface quality and tool wear. Substantial process knowledge and precision engineering are therefore involved in the honing stone specifications. The test results demonstrate the differences of honing tool specifications and their performance in honing the high-strength steel 100Cr6 and current coating systems in automotive industry with similar material properties. In addition to the G-Factor, the grain utilisation factor (U-Factor) is introduced to quantify the honing stone wear. This characteristic value allows for a broader comparability and can be used to better determine the effects of the honing stone specification on the actual diamond abrasive wear. Thus, reduced concentration caused improved utilisation of the grains.

Due to minor differences in surface roughness of the honing stone specifications examined, it is concluded that a single prehoning and a single intermediate honing step is sufficient for an efficient process chain. In contrast, for finishing a significant reduction in surface roughness was achieved. Due to these roughness reduction, two honing steps should be applied for the highest surface qualities. Considering the honing stone wear specifications with monocrystalline diamond grains achieve a higher grain utilization and show less wear marks. Honing stones with multicrystalline grains show a grain wear exclusively by splintering. In addition, grain interaction and grain dispersion quality in the bond were identified as relevant factors. Here, it could be derived that a slightly reduced grain concentration enables a higher grain utilisation by reduced interaction with particles in the gap between honing stone and surface. The interaction and collisions of bonded grains with particles and lose grains and grain splinters induce premature wear in form of splintering or breaking out and thus increase the wear rate. An improvement in honing stone specifications in terms of reduced wear rates is accompanied by improved dispersion of grains and tuned grain concentration. With these results, the composition of honing stones and the planning of honing processes can be improved for the highest machining performance combined with the best grain utilisation.

**Author Contributions:** Conceptualization, methodology, validation, formal analysis, investigation, data curation, writing—original draft preparation, visualization, G.M.; writing—review and editing, supervision, K.D. All authors have read and agreed to the published version of the manuscript.

**Funding:** This research received no external funding.

**Data Availability Statement:** The research data is available on request.

**Acknowledgments:** We acknowledge support by the German Research Foundation and the Open Access Publication Funds of Technische Universität Braunschweig. The authors would like to express their thanks to Christian Homann for his support in experimental validation.

**Conflicts of Interest:** The authors declare no conflict of interest.

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
