# Peer review of "Performance and Wear of Diamond Honing Stones"

_machines, doi:10.3390/machines11040502_

Round 1

Reviewer 1 Report

The subject of the paper is very intersting. However, he documents has some concerns:

-      -    The introduction should consider other authors’ results about wear of the honing stones and wear of grinding wheels. Too few references were considered in this section.

-       -   The novelty of the paper should be highlighted in the last paragraph of the Introduction section.

-        -  Were the results in Figure 4 obtained by the authors of the paper? Which machine was employed? Which machining conditions were employed?

-       -   A section including the detailed description of all the experimental methods is required

-        -  The results in Table 3 and Table 4 should be discussed in more detail and compared to other authors’ results. In general, section 4 requires more discussion, and to mention different bibliographic references.

Other minor concerns are as follows:

-        -  Section 1.A bibliographic reference is required for the sentence “With a total of 65 %, industrial diamond is the most widely applied cutting material in honing”.

-       -   Section 2. Line 52. “Occur” should be replaced with “Occurs”.

-        -  Section 2. Line 94. “To a large extend” should be “To a large extent”.

Author Response

1. The introduction should consider other authors’ results about wear of the honing stones and wear of grinding wheels. Too few references were considered in this section.
- Paragraphs on the wear of honing stones and grinding wheels by other authors have been added to the introduction with additional references.
2. The novelty of the paper should be highlighted in the last paragraph of the Introduction section.
- The novelty of the article has been added in the context of further literature and its analysis in the introduction.
3. Were the results in Figure 4 obtained by the authors of the paper? Which machine was employed? Which machining conditions were employed? The images and data from Fig. 4 were developed within this study in the same test set-up as the other test results.
- The machining conditions are summarised above the image, as for all images with test results. A remark has been added to the picture description.
4. A section including the detailed description of all the experimental methods is required.
- An additional section “4. Experimental setup and machining conditions” has been added.
5. The results in Table 3 and Table 4 should be discussed in more detail and compared to other authors’ results. In general, section 4 requires more discussion,and to mention different bibliographic references.
- For the surface roughness in Table 3 additional references are provided in Chapter 5. The focus of the analysis is on cutting performance and wear. The surface quality, roughness and texture are part of another article published in the future. For this reason, the surface will not be discussed in more detail.
6. Other minor concerns are as follows:
Section 1.A bibliographic reference is required for the sentence “With a total of 65 %, industrial diamond is the most widely applied cutting material in honing”. Reference is added.
Section 2. Line 52. “Occur” should be replaced with “Occurs”. Changed
Section 2. Line 94. “To a large extend” should be “To a large extent”. Changed

Reviewer 2 Report

Notes in the attachment.

Author Response

1. The authors did not specify the exact purpose of the study.
- In the abstract and introduction, the aim of the study is highlighted.
2. No description schematic representation of the honing head.
- In the experiments, a test rig for external circular honing was used as described in section 4. The machine operates with a single oscillating honing stone. The rotational movement and the contact pressure are applied to the workpiece. A more detailed description and a picture of the test rig can be found in [2, 8].
3. No description of working conditions (axial speed, rotation speed, number of stones, abrasive grit type, grain size, bond type).
- The machining parameters are included in each image or diagram. For better comprehensibility, an additional chapter „4. Experimental setup and machining conditions” with a detailed description of all conditions is added.
4. Which standard is the surface roughness criteria based on? On the ISO 13565-2?
- Surface roughness was measured tactile according to DIN EN ISO 27178. Based on the surface profile characteristics the robust profile filter Gaussian regression filter according to DIN EN ISO 16610-31 was used. The material content curve (Abbot) was calculated as per DIN EN ISO 13565-2. The applied standards have been added to the manuscript.
5. Why were the following parameters: Rk, Rpk, Rvk not used for the assessment of roughness?
- The values for Rk, Rpk, Rvk are given in Table 3 and discussed afterwards for the used honing stone specifications. The subsequent analysis of the process performance is carried out on the basis of the average roughness depth Rz, in order to ensure a more clearly arranged presentation in the diagrams and images. The surface quality, roughness and texture are part of another article published in the future. For this reason, the surface will not be discussed in more detail.
6. Honing texture of cylinder liners honed. Please show pictures of these surfaces.
- The surface quality, roughness and texture are part of another article published in the future. A detailed analysis of the surfaces would enlarge this article too much.
7. Evolution of stones wear and honing ratio. Please describe this phenomenon using a graph.
- In the experiments, no variations in honing stone wear and ratio could be detected with increasing machining volume. In the literature a constant wear rate is stated once the initial run-in of the honing stones is finished. This is referred to as the self-sharpening effect of honing stones. For this reason, the evolution of stone wear is not discussed in this article.
8. No description of the mechanical properties of 100Cr6 steel.
- The 100Cr6 material used is heat-treated to 364 ± 12 HV 0.3. This information is provided in chapter 4.
9. No photos of the topography of the machined surface
- The surface quality, roughness and texture are part of another article published in the future. A detailed analysis of the surfaces would enlarge this article too much.

Round 2

Reviewer 1 Report

The paper has improved significantly. However, the conclusions are too generic and should be related to the main findings of the paper.

Author Response

Review 1: 1. The paper has improved significantly. However, the conclusions are too generic and should be related to the main findings of the paper. Further improvement of the conclusion is included in the new article.

Reviewer 2 Report

The authors have made corrections to the article. I believe the article can be published.

Author Response

Review 2: 1. The authors have made corrections to the article. I believe the article can be published. Thank you very much for this assessment.
